# The Exposome Approach to Decipher the Role of Multiple Environmental and Lifestyle Determinants in Asthma

**DOI:** 10.3390/ijerph18031138

**Published:** 2021-01-28

**Authors:** Alicia Guillien, Solène Cadiou, Rémy Slama, Valérie Siroux

**Affiliations:** Inserm, CNRS, Team of Environmental Epidemiology Applied to Reproduction and Respiratory Health, IAB (Institute for Advanced Biosciences), University Grenoble Alpes, 38000 Grenoble, France; solene.cadiou@polytechnique.org (S.C.); remy.slama@univ-grenoble-alpes.fr (R.S.); valerie.siroux@univ-grenoble-alpes.fr (V.S.)

**Keywords:** asthma, exposome, epidemiology, statistical methods

## Abstract

Asthma is a widespread respiratory disease caused by complex contribution from genetic, environmental and behavioral factors. For several decades, its sensitivity to environmental factors has been investigated in single exposure (or single family of exposures) studies, which might be a narrow approach to tackle the etiology of such a complex multifactorial disease. The emergence of the exposome concept, introduced by C. Wild (2005), offers an alternative to address exposure–health associations. After presenting an overview of the exposome concept, we discuss different statistical approaches used to study the exposome–health associations and review recent studies linking multiple families of exposures to asthma-related outcomes. The few studies published so far on the association between the exposome and asthma-related outcomes showed differences in terms of study design, population, exposome definition and statistical methods used, making their results difficult to compare. Regarding statistical methods, most studies applied successively univariate (Exposome-Wide Association Study (ExWAS)) and multivariate (adjusted for co-exposures) (e.g., Deletion–Substitution–Addition (DSA) algorithm) regression-based models. This latest approach makes it possible to assess associations between a large set of exposures and asthma outcomes. However, it cannot address complex interactions (i.e., of order ≥3) or mixture effects. Other approaches like cluster-based analyses, that lead to the identification of specific profiles of exposure at risk for the studied health-outcome, or mediation analyses, that allow the integration of information from intermediate biological layers, could offer a new avenue in the understanding of the environment–asthma association. European projects focusing on the exposome research have recently been launched and should provide new results to help fill the gap that currently exists in our understanding of the effect of environment on respiratory health.

## 1. Introduction

Asthma is a heterogeneous chronic respiratory disease characterized by an inflammation of the airways and which manifests by variable respiratory symptoms (wheeze, shortness of breath, chest tightness and/or cough) and variable expiratory airflow limitation [1]. Asthma affects approximately 300 million children and adults worldwide [2]. The prevalence of asthma has dramatically increased over the last decades [3]. The huge research efforts in identifying the causes of asthma led to the identification of genetic (such as the 17q21 ORMDL3/GSDML region for early childhood onset asthma), environmental (such as urban vs. rural area) and lifestyle risk factors (such as tobacco smoking). It also highlighted the complex etiology of this multifactorial disease, e.g., by the identification of specific windows of susceptibility and complex gene-by-environment interactions [4]. The exposome concept, introduced in the recent years to complement the genome for a better understanding of the development of complex diseases [5], offers new avenues in environmental epidemiology. In this review, the main objective was to present how the new methodological framework represented by the exposome has been applied to asthma research to date. After presenting an overview of the concept of exposome, we will review different statistical approaches to study the exposome–health associations. Finally, recent studies linking multiple families of exposures to asthma-related outcomes will be discussed.

## 2. The Emergence of the Exposome Concept

In the twentieth century, the global average life expectancy almost doubled, and the distribution of the causes of death has changed, in particular in industrialized countries, with a shift in causes of death from infectious diseases to chronic diseases. These changes, referred as the epidemiological transition, are the result of a combination of several factors, including demographic changes (ageing population), modern medicine practice (e.g., antibiotics, vaccines), and changes in risk factors in relation to a more complex environment [6]. Indeed, the industrialization led to a profound transformation of our society, which resulted in changes to our environment (e.g., atmospheric pollution, chemical pollutants with the development of synthesis chemistry…) and lifestyles (e.g., change in dietary patterns, decreased physical activity, …). The chemical revolution (mid 20th-century) led to a huge increase in chemical production, from 1 million tons/year in 1930 to 1 billion tons/year in 1990 [7], and in the number of chemical products (more than 23,000 substances have been registered in the EU, according to the Reach Procedure of the European Union [8].

Regarding asthma, the strong increase in disease prevalence in less than 50 years in industrialized countries, where it doubled and sometimes even tripled [1,9], suggests the role of environmental and behavioral factors for which a change in exposure has occurred in this time window. In epidemiology, the usual approach to address environmental risk factors on health has long been to consider each factor separately (together with the potential confounders of its relation to health), or by focusing on a family of exposure at a time. Following this approach, a large range of environmental factors and behaviors have been identified or are suspected to be involved in the development of asthma and allergic diseases, as reported in recent reviews [10,11]. One of the most documented risk factors for asthma is tobacco smoking, with effects whatever the window of exposure (in utero maternal smoking, parental smoking during childhood and active smoking in adulthood) [12,13]. Early life is a landmark period for the development of asthma: early exposure to respiratory viral infection [14,15], aeroallergens [16], outdoor air pollution [17] and pets are all associated with increased risk of childhood asthma while early exposure to farm animals [18] is associated with a decreased risk. In adulthood, these factors are still associated with asthma-related outcomes, such as exacerbations or severity of asthma. It is now well recognized that long-term exposure to outdoor air pollution is associated with onset of childhood asthma [19] and that acute exposure can make asthma symptoms worse and trigger asthma attacks both in children and adults with asthma [20,21]. In adulthood, occupational exposures to 250 agents may induce work-exacerbated asthma in individuals with pre-existing asthma or occupational asthma in individuals without pre-existing asthma [22]. Regarding lifestyle factors, evidence showed that obesity and poor diet were associated with an increased incidence of asthma in adults [23], while a healthier diet was associated with fewer asthma symptoms and greater asthma control [24]. Until now, this knowledge has led to the development of few untargeted preventive strategies [25]. The first one was the ban of smoking in public places. According to a Cochrane review performed on 12 studies, seven reported that the smoking ban was associated with a significant reduction in hospital admission rates for asthma in children and adults [26]. The use of vitamin D3 supplementation during pregnancy was also tested on a mother–child cohort from Denmark [27]. This double-blind randomized controlled trial showed a difference, although not statistically significant, in the incidence of persistent pre-school wheeze and a reduced number of pre-school wheezing episodes, all not sustained into school age. Nevertheless, these measures were not sufficient to reduce the global asthma incidence. The factor-by-factor analytical approach might be of limited value if exposures mostly affect health through complex synergistic effects with other exposures or lifestyle factors. A more global approach, considering the environment in the broad sense, could help further decipher the etiology of this complex chronic disease.

The exposome concept was first coined by C. Wild in 2005 to encompass “the totality of human environmental exposures from conception onwards, complementing the genome”. It was introduced to underline the imbalance between the data often used to characterize the environment in etiological studies, relying mainly on questionnaires, and the more finely and extensively covered genetic information, since the advent of genome-wide approaches [5]. In 2012, Wild [28] defined a general exposome framework, composed of three domains, to differentiate types of exposures: 1) a general external environment, referring to exposures shared at the community level, (e.g., air pollution, social factors, urban–rural environment,… ) and mainly assessed through geographic information system-based models; 2) a specific external environment, referring to exposures that are specific to each individual (e.g., diet, physical activity, tobacco, occupation, …), usually assessed through questionnaires or personal censors; and 3) an internal environment, referring to biological measures of toxicants entering the body but also the biological and toxicological perturbations related to external exposures’ effects in the body (e.g., metabolic factors, microbiota, …) mainly assessed through biological chemical measures, including high-output techniques [29] (Figure 1.). Miller and Jones [30] expanded the exposome concept to “the summation and integration of external forces acting upon our genome throughout our lifespan”, and therefore explicitly added behaviors, as well as the body’s response and endogenous processes changing in response to environmental exposures. Nevertheless, efforts to widely assess the exposome should not be made at the expense of the accuracy of exposure measurements, since measurement errors in exposures have major impacts on the performance of exposome–health association studies [31].

## 3. Exposome-Health Associations in Practice: Statistical Approaches

Exposome studies imply collection of a large number of exposures. This can be done relying on different methods of assessment (e.g., self-reported questionnaire, exposure biomarkers, geographic information system-based (GIS) models, personal sensors, …), for different time windows (pre-natal, early postnatal, during childhood, adolescence, adulthood), and, for external factors, with different locations (home, school, work) and spatial resolutions (e.g., urban indicators measured for various buffers (100, 300, 500 m)). From a methodological point of view, this large number of variables (possibly larger than the size of the study population) raises issues in terms of statistical power and false discovery rate [32]. Indeed, the multiplicity of tests implies the rise of the alpha risk, and methods developed to correct the p-value of an association for multiple hypothesis testing [33,34,35] lead to a decreased statistical power. Therefore, exposome–health association studies deserve a sufficient sample size to achieve adequate statistical power to detect associations of low to moderate associations sizes, as expected for most exposures [36,37]. Several other statistical challenges specifically linked to exposome studies have to be taken into account, such as the increased false discovery rate related to the high level of correlation between exposures and the difficulty to consider “mixture” effects [38,39]. Until now, no consensus establishing which statistical methods are to be used in exposome–health association studies has been reached [40]. However, some simulations have allowed the comparison of the performance of various methods in the exposome research context under some specific settings. For example, simulation studies compared i) the efficiency of various regression-based approaches in terms of false positive rate and sensitivity, with and without interactions between exposures [32,39]; ii) the performance of variable selection models in case-control studies [41]; iii) the performance of variables and function selection methods in the case of nonlinear effects of correlated exposures [40]; and iv) methods to correct for classical-type exposure measurement error [31]. Using the findings of these studies and a review of the literature, we summarized in Table 1 the strengths and weaknesses of the main statistical approaches used in exposome studies in the field of respiratory health.

**Table 1 ijerph-18-01138-t001:** Main statistical methods used in exposome-health association studies.

Type of Analysis	Examples of Methods	Strengths	Weaknesses	Reference of the Method	Use of the Method in Exposome or Asthma Field
Single-exposure regression-based method	Exposome-Wide Association Study (ExWAS)	Standardized method High sensitivity to identify true predictors Simple interpretation Easy to summarize the results in a figure (e.g., volcano plot)	Interaction between exposures is not tested Results do not account for confounding effect by co-exposures High false discovery rate	Patel et al., 2010 [42]	Sbihi et al., 2017 [43]; North et al., 2017 [44]; Lepeule et al., 2018 [45]; Agier et al., 2019 [46]; Vrijheid et al., 2020 [47]; Agier et al., 2020 [48]; Warembourg et al., 2019 [49]; Nieuwenhuijsen et al., 2019 [50]; Granum et al. [51]
Multiple-exposures regression-based methods	Deletion–Substitution–Addition (DSA) algorithm	All exposure variables are considered in a unique model with possibility to include interactions The selected model is able to account for confounding effect by co-exposures Low false discovery proportion to identify true predictors	Moderate sensitivity to identify true predictors Instability Time-consuming and thus not adapted for exposome of more than a few hundred variables	Sinisi and van der Laan 2004 [52]	Agier et al., 2019 [46]; Vrijheid et al., 2020 [47]; Agier et al., 2020 [48]; Warembourg et al., 2019 [49]; Nieuwenhuijsen et al., 2019 [50] Granum et al. [51]
	Elastic Net (ENET) and Least Absolute Shrinkage and Selection Operator (LASSO)	Able to deal with correlated variables The selected model is able to account for confounding effect by co-exposures Good prediction performance	Moderate sensitivity to identify true predictors Instability	Zou and Hastie 2005 [53]; Tibshirani 1996 [54]	Pries et al., 2019 [55]; Cowell et al., 2019 [56]
	Weighted Quantile Sum (WQS) regression	Able to deal with multicollinearity The use of quantiles reduces the impact of outliers	Not able to consider categorical exposures All exposures must be associated with the outcome in the same direction (i.e., all protective or all risks factors)	Carrico et al., 2015 [57]	-
Supervised clustering approaches	Latent Class Analysis (LCA)	Suitable for longitudinal data (Latent Transition Analysis [58]) Able to consider the outcome in a supervised approach	Not able to deal with continuous exposures Model requires low correlation between variables Interpretation of results may be difficult in case of large number of clusters Limited dimension of the exposome (in relation to the sample size)	Goodman et al., 1974 [59]	Buck Louis et al., 2019 [60]; Harmouche-Karaki et al., 2019 [61]
	Bayesian Profile Regression (BPR)	Consider all exposure variables in a unique model Able to determine the number of clusters minimizing the least-squared distance to the probability matrix Able to deal with combined continuous and categorical variables	Computing time Interpretation of results may be difficult in case of large number of clusters Unstable method	Molitor et al., 2020 [62]	Berger et al., 2020 [63]; Belloni et al., 2020 [64]
Analysis accounting for the hierarchical structure of the data	Meet-in-the-Middle (MITM)	Considers the hierarchical layers in the exposome and the causal link between them to better document the causality in exposome–health associations	Needs an a priori selection of intermediate layers	Chadeau-Hyam M et al., 2011 [65].	Vineis et al., 2020 [66]; Jeong et al., 2018 [67]; Cadiou et al., 2020 [68]
	Bayesian Kernel Machine Regression (BKMR)	Use of a smooth kernel function able to deal with non-monotonic exposure-outcome relationship Able to deal with a priori knowledge about group of exposures Able to deal with multicollinearity	Not able to deal with categorical outcomes The hierarchical variable selection option can select only one variable per group	Bobb et al., 2015 [69]	Berger et al., 2020 [63]

### 3.1. Single-Exposure Regression-Based Models

The simplest approach to study the exposome–health association consists of describing single-exposure associations. In that order, one regression model is fitted for each exposure variable separately, and eventually adjusted for confounders. These models could also include random effects, using mixed models, when data are hierarchically structured (e.g., multicenter studies, family-based studies). This approach has been standardized and is called Exposome-Wide Association Study (ExWAS), by analogy with the Genome-Wide Association Study in the genetic field. The ExWAS has first been used in a pilot exposome study aimed at exploring the role of environment on type 2 diabetes mellitus [42]. The main weakness of this method is that it is not able to account for confounding by co-exposures nor for interactions between exposures, the so-called “mixture” effect, while we know that individuals are continuously exposed to combined exposures in their daily life. Nevertheless, this agnostic method has the advantage of providing results that are easy to interpret, summarize in a figure (e.g., a volcano plot) and compare with the results of the literature. The performance of this approach, in terms of sensitivity and specificity, has been addressed in a simulation study that compared six linear regression-based methods to assess exposome–health associations in a realistic setting [32]. Among them, the ExWAS method achieved the best sensitivity (mean sensitivity (95%CI): 0.96 (0.90–0.98)) to identify true predictors among the 237 simulated exposure variables but had a very high false discovery proportion (FDP) even after correction for multiple testing (mean FDP (95% CI), 0.86 (0.67–0.93)).

### 3.2. Multi-Exposure Regression-Based Models

To go a step further from single-exposure analyses, multivariate analyses allow correction for cofounding by co-exposures of multiple exposures. For example, an extension of the ExWAS consists of fitting a multivariate linear/logistic regression (ExWAS-MLR) from all the exposures reaching a p-value below 0.10 (or below 0.20 in some cases) [46]. Moreover, in a few exposome studies [46,47,48,49,50,51], the iterative Deletion–Substitution–Addition (DSA) algorithm has been applied. This algorithm switches between deletion, substitution and addition of variables in order to identify the multivariate model minimizing the root mean square error (RMSE) using x-fold (generally five) cross-validated data [52]. This method has the advantage of adjusting for the selected co-exposures, to allow addition of nonlinear terms and inclusion of interaction terms between exposures. A simulation study comparing six regression methods in a two-way interactions setting showed that this DSA approach reached the lowest FDP (mean FDP (95% CI): 0.28 (0.21–0.33)) while keeping mean sensitivity at 0.73 (0.65–0.80) [32]. Nevertheless, this method is time-consuming and suffers from high instability due to the cross-validation procedure. Indeed, to achieve a stable selection, exposome studies which have used DSA so far had to add a stabilization step, considering only the exposures selected in an arbitrary-set fraction of repeated runs, a protocol which could possibly increase the FDP [70]. Other multivariate analyses appropriate for exposome studies are the penalized regression models, such as the Elastic Net (ENET) and the Least Absolute Shrinkage and Selection Operator (LASSO) methods [53,54]. These methods estimate regression coefficients for all exposure variables with the least informative predictors attributed to estimates close to zero. In two simulation studies, ENET performed better in term of specificity than ExWAS and DSA but at the cost of a reduced sensitivity [32,41]. Finally, in the context of environmental chemicals, Weighted Quantile Sum (WQS) regression has been developed, with the aim of dealing with multicollinearity between exposures [57]. With this method, the weighted sum of the quantiles of each exposure chemical is considered in a regression model. The use of quantile of exposures has the advantage to not be influenced by potential outliers but makes the model unable to deal with categorical exposures. Moreover, one of the constraints imposed on the weights associated with exposures is to be between zero and one: thus, the effects of all exposures are supposed to act in the same direction (the WQS cannot consider “protective” and “risk” factors in the same model). Recently, this method has been extended by adding a causal inference method, known as g-computation, to estimate the joint effect of all exposures in a mixture without assumption on directional homogeneity [71]. Finally, the g-computation allows for non-linearity and non-additivity of the effects of individual exposures and the mixture as a whole.

### 3.3. Cluster-Based Analyses Relating Profiles of Exposure to Health Outcome

Another way to tackle exposome–health associations is to identify groups of individuals sharing the same exposures pattern and outcome level by performing supervised cluster-based analysis. The general approach relies on two principles: the within-cluster homogeneity (two individuals from the same cluster should have similar exposures and outcome) and the between-cluster separability (two individuals from two different clusters should have different exposures and outcome). This approach, that takes into account all exposures simultaneously, is able to account for the additive and mixture effects of the exposures; thus, it is expected to be more representative of the mode of exposure in real life. Overall, clustering methods take into account correlations between exposure variables and reduce the size of the exposome (a set of hundreds of exposures is summarized in a finite number of clusters, generally lower than 10). Thus, this approach has the major advantage of dealing with the multiple tests issue since it allows the comparison of these clusters of individuals in terms of health outcome in a single test experiment. By comparison, the ExWAS analysis performed on an exposome composed of *k* exposures would require *k* tests. Moreover, some clustering approaches also include a selection of variables step. Currently, several clustering methods exist, including K-means, hierarchical clustering, and model-based approaches (e.g., Latent Class Analysis (LCA) or Bayesian Profile Regression (BPR)) [59,62]). Compared to other clustering methods, BPR is able to deal with intra-class dependencies, and therefore allows the inclusion of correlated exposures and to deal with both continuous and categorical variables. However, this method might suffer from instability, in particular with a wide exposome. To limit this instability issue, a recent study proposed to apply the supervised BPR in a restricted set of exposures selected from the ExWAS results (detailed below) [72].

### 3.4. Analyses Accounting for the Hierarchical Structure of the Data

The exposome can be seen as being composed of several layers of data, as illustrated in Figure 1, and this hierarchical structure of the data could be integrated into the statistical analysis to account for the possible causal links between the different layers. As an example, because methylation is considered as a biological pathway between environmental exposures and health, information from methylome data can be usefully integrated in an exposome–heath association study. Among the different existing approaches, the meet-in-the-middle approach consists of studying the association between intermediate layers, usually biomarkers, with external exposures (the exposome) and subsequently with a health outcome. Therefore, this approach attempts to account for the biological structuration of the different layers by identifying putative mediators [65]. Recently, a tailored Meet-In-The-Middle (MITM) approach [68] has been proposed to overcome one of the main issues of exposome studies, that is the high false-positive rate [32]. The approach proposed by Cadiou S et al. relies on a four-step approach: (1) selection of genes involved in biological pathways a priori relevant for the health outcome; (2) association study between Cytosine-Phosphate-Guanine (CpGs) in genes identified in step 1 and the outcome; (3) association study between the exposome and the CpG sites identified in step 2 (adjusted for the outcome); (4) association study between the reduced exposome identified in step 3 and the health outcome. The authors of this targeted MITM approach hypothesized that it could lead to a higher specificity in the identification of true predictors compared to the agnostic approach (e.g., ExWAS), although a simulation study would be required to validate this hypothesis.

Another way to consider exposome data is to group exposures by family (e.g., phenols, phthalates, air pollutants), which could be seen as a cluster approach relying on a priori information. For example, Bayesian kernel machine regression (BKMR) has been proposed to estimate the health effects of combined exposures (mixtures) by integrating information of data structure [69]. In BKMR the health outcome is regressed on a flexible function of the combined exposures (specified with a kernel function). To account for correlated exposures, a hierarchical variable selection can be included in the modeling approach to incorporate a priori knowledge on the structure of the exposome (e.g., exposure families). At this time, this method is available only for continuous outcomes.

All these methods present strengths and weaknesses and none of them can be considered as more efficient in all cases. The choice of the statistical method should be made according to the design of the study, the type of outcome, the type of exposome data, the sample size and to the main objective of the study. Therefore, the results of such studies should be interpreted considering the characteristics of the statistical method used.

## 4. Exposome and Asthma: Literature Review

In past decades, many studies exploring the role of various individual environmental factors on asthma characteristics have been published, and summarized in recent literature reviews [10,11]. Here, in order to address how the new methodological framework represented by the exposome has been applied to asthma research, we conducted a PubMed search for journal articles published up to November 30, 2020 with the search terms “(respiratory OR lung function OR asthma OR allergy) AND “exposome”. As there is no rigorous definition for exposome when it comes to studying exposome–health association in epidemiology, we a priori arbitrarily defined exposome studies by studies investigating simultaneously at least two distinct families of exposures and, overall, at least ten different exposure variables, acknowledging that this could be debated. The PubMed search identified 141 full texts, of which 89 were not original articles but commentaries or reviews. Among the remaining 52 manuscripts, only four original studies investigated the link between at least two families of exposures and 10 exposure variables, and asthma or respiratory outcomes while the 48 others did not consider large range of exposures or did not consider asthma-related outcomes.

Among the four original exposome studies identified, three performed single-exposure analyses, followed by multiple-exposures analyses, eventually including a variables selection step. In contrast, the most recent study applied a two-step approach consisting of single-exposure analyses (ExWAS) followed by a supervised cluster analysis on a restricted set of relevant exposures from ExWAS results.

The study that uses for the first time the term “exposome” in the field of respiratory epidemiology was based on the Kingston Allergy Birth Cohort and aimed at exploring the role of exposome on respiratory symptoms of 235 children [44]. The exposome was assessed through 21 prenatal and 17 postnatal exposures including a general external exposome (socioeconomic status (SES), rural or urban residence), specific external exposome (cigarette smoke, breastfeeding, mold or dampness), and “internal exposome” (gestational age and some clinical parameters). Univariate Cox proportional hazard models, not corrected for multiple testing, identified two prenatal (SES and prenatal smoke) and five postnatal exposures (SES, air freshener, candles or incense, indoor mold and postnatal smoke) positively associated with parental report of respiratory symptoms at 2 years of age, with hazard ratios (HR) ranging from 1.85 (1.09–3.13) for prenatal SES to 3.26 (1.56–6.78) for postnatal smoke. Breastfeeding for at least 6 months was negatively associated with the child outcome (HR (95% Confidence Interval, CI): 0.41 (0.23–0.71)).

More recently, the European Human Early-Life Exposome (HELIX) project combined six European longitudinal birth cohorts to explore the role of exposome on various health outcomes [46,51,73] using an ExWAS and the DSA algorithm. The exposome was composed of 85 prenatal and 125 postnatal exposures covering 17 exposure families, including chemicals. Among 1033 children with validated spirometry data (median (inter quartile range, IQR) age = 8.1 (6.5–9.0) years), the forced expiratory volume in 1 s (FEV_1_) was positively associated with three prenatal exposures (two perfluorinated alkylated substances’ biomarkers levels and the distance of the residence to the nearest road during pregnancy). On the contrary, FEV_1_ was negatively associated with nine postnatal exposures (copper, ethyl-paraben, five phthalate metabolites internal levels, house crowding and facility density around schools) in ExWAS adjusted for confounders [46]. None of these associations remained statistically significant after correction for multiple testing and no exposure was selected with the DSA algorithm. Furthermore, based on the Helix cohorts, an analysis considered the role of the exposome (composed of 90 prenatal and 107 childhood exposures) on allergy-related outcomes [51]. The ExWAS analysis identified two prenatal (inverse distance of the residence to nearest road and one phthalate) and three childhood exposures (population density, cadmium biomarker and one phenol biomarker) positively associated with rhinitis in the 1270 6–11-year-old considered children. Moreover, one prenatal (particulate matter absorbance) and three childhood exposures (cat at home, blood molybdenum and perfluorooctane sulfonate levels) were negatively associated with rhinitis. None of these nine associations remained statistically significant after correction for multiple testing and only the three prenatal exposures were selected in the additional DSA analysis. These three first studies should be commended as they provided a first step towards the exposome approach in respiratory research by systematically reporting the association of a large set of exposures with asthma-related outcomes using a standardized protocol within each study. They confirmed some previous findings (e.g., prenatal smoke, breastfeeding) and identified some suspected factors (e.g., phthalate metabolites) for which further investigations would be required. Nevertheless, these first studies did not attempt to address the effect of combined exposures. When dealing with a multifactorial disease or health parameter, such as asthma, allergic diseases and lung function, comprehensive statistical approaches able to jointly consider a large set of exposures are needed.

Finally, the fourth study identified relied on an older French population, the Epidemiological Study on the Genetics and Environment of Asthma (EGEA) [74]. In this study, the role of exposome, assessed by 53 lifestyle/environmental exposures from 17 families of exposures, on the forced expiratory volume in 1s (FEV_1_) of 599 adults with asthma was investigated [72]. The first step of the statistical analysis consisted of an exposome dimension reduction by selecting exposures variables which showed a trend for an association with FEV_1_ (p-value below 0.20 in the ExWAS analysis). Then, a supervised BPR method was applied by considering the selected exposure variables in step 1 and FEV_1_, in order to identify clusters of individuals sharing a similar profile of exposures and a similar level of lung function. The study identified three clusters including one cluster of 30 individuals showing the lowest mean ± SD FEV_1_ (79% ± 21 vs. 90% ± 19 and 93% ± 16) and characterized by a specific exposure pattern (heavy smoking, poor diet, higher outdoor humidity and proximity to traffic). Interestingly, this study identified a specific profile of joint lifestyle and environmental factors associated with a low FEV_1_ in adults with asthma while none of the exposures revealed significant association when considered independently in the ExWAS. The differences in the results observed between these two steps using different statistical methods (single exposure regression-based model, ExWAS, and cluster-based analysis, the BPR) support the hypothesis that comprehensive statistical approaches could be useful to address the effects of the environment in complex multifactorial health parameters.

These four studies constitute the very first efforts to address the respiratory health environmental determinants via an exposome approach. This approach is still in its infancy, but we can expect that the literature will be expanded with original studies based on detailed exposome [75] and comprehensive approaches.

## 5. Conclusions

Asthma is a widespread multifactorial disease, which deserves a comprehensive approach to better understand its etiology and development. Although most of previous studies in environmental epidemiology focused on a single exposure (or single exposure family), with the recent emergence of the exposome concept, several studies and European projects have started to assess the effect of multiple exposures on respiratory health. These studies are expected to contribute to a better understanding of the associations between the environment and health by using various holistic approaches. Although the first association studies between the exposome and asthma-related outcomes conducted so far mainly rely on the ExWAS method for successive single-exposure analysis and the DSA algorithm for multi-exposures analysis [46,48,49,50,51,76], further studies on larger sample size should attempt to apply more comprehensive statistical approaches, either able to account for the hierarchical structure of the multiple layers of the exposome or to account for the possible mixture effects in order to be more consistent with the complex structure of exposure data.

## Figures and Tables

**Figure 1 ijerph-18-01138-f001:**
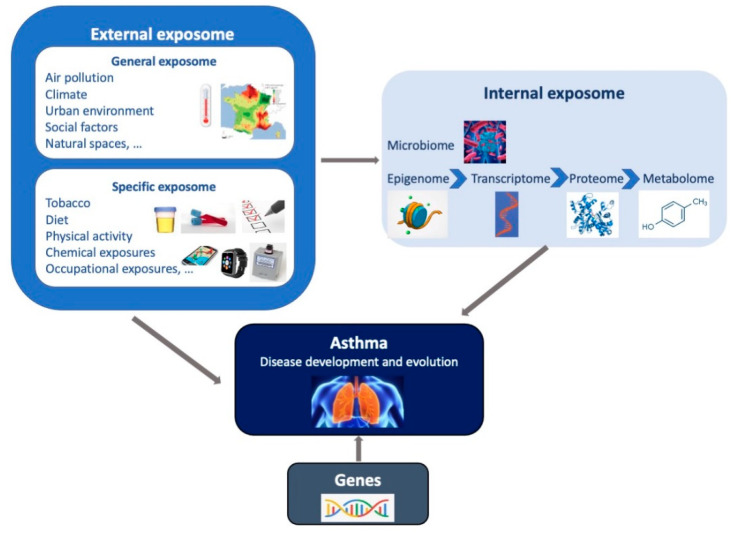
Representation of the exposome concept. The exposome represents the totality of human environmental exposures from conception onwards, complementing the genome. The exposome is composed of three domains (general external, specific external and internal) which may interact together and with genetic factors, to impact on the asthma development and on the manifestation of the disease among individuals with asthma.

## Data Availability

Not applicable.

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
