# Peer review of "The Exposome Approach to Decipher the Role of Multiple Environmental and Lifestyle Determinants in Asthma"

_ijerph, 2021, doi:10.3390/ijerph18031138_

Round 1

Reviewer 1 Report

This is a very nice review article about Asthma towards the exposome approach. However, there is some missing information that should be added in this review article. 

Line 75: The author mentioned, "A large range of environmental factors and behaviors have been identified or are suspected to be involved in the development of asthma and allergic diseases (e.g. passive and active smoking, respiratory viral infections, aeroallergens, occupational exposures, diet, pets, living on a farm, atmospheric pollution)" 

Could the author be more specific on each factor in the brackets? Explain each factor in a few sentences with important references. 

Line 80: The author mentioned here, (e.g. smoking ban in public places, vitamin D3 supplementation during pregnancy)

Please, explain each factor in a few sentences with important references.

Line 91: The author mentioned here,

In 2012, Wild [11] identified three domains within the exposome 1)a general external environment, referring to climate factors, social factors, urban-rural environment, etc.; 2) a specific external environment, referring to diet, physical activity, tobacco, occupation, etc.; and 3) an internal environment, referring to biological factors, such as metabolic factors or microbiota, and to some extent the biological and toxicological manifestations of exposures in the body (see Figure). 

Please, rephrase the paragraph for a better understanding. The author should expand this paragraph according to each factor. For example, what does climate factors associate with Asthma? along with the following factors: social factors, urban-rural environment, diet, physical activity, tobacco, occupation, metabolic factors, and microbiota. 

Figure: Please, describe the figure legend in detail, not just one sentence.

In the section of the internal exposome, the author mentioned many factors here but never explain them in the text. It would be better to explain more in the text in order to match the figure. 

The table is nicely prepared and matched with the text. Please, don't change the table.

It would make more sense to show the topic of "Exposome and asthma: literature review" after explaining the meaning of exposome and then show the table. 

Reviewer 2 Report

Abstract-  Many run on sentences makes this abstract in particular but the manuscript in general really hard to read and understand. Please cut these longer sentences into shorter sentences with more appropriate punctuation.  Might benefit from English editing. Line 28 reads funny (…but does not allow [one] to address complex interactions). The issue with this sentence is observed in multiple places throughout.

A good overview of exposome is included. Table 1 is easy to follow and has value. “Strengths” is spelled wrong on column title. Some methods include acronyms (ENET, WQS, etc.) and some do not (LCA, BPR, etc.). Its best to be consistent and either include them for all or not. Given you use them in the text, they should be included in the table.

Search term of exposome really limits the pool as this is a very recently used term. It looks like your table has only publications from 2017 to present. The authors point out that this term omits publications that might have value. Then why do they not expand their search? Might the search be expanded or a better explanation be included to why the search was limited? It seems stating that earlier studies commonly used univariate/multivariate approaches...assuming this is true.

Four publications are finally reviewed focusing on results but they were not compared to each other. Might they have different outcomes if they had used another statistical test? The point of this paper seems to be to highlight that focusing on the exposome in asthma studies has great value despite the accompanying statistical challenges. The discussion would benefit from greater synthesis of the publications described and these publications should more clearly make the point that newer analytical approaches could be beneficial. Sample size is mentioned as a side bar in the conclusion but not really discussed before. The ending of the conclusion lines 353-355 seems not to match the “story” of the article. It is not that it is not worth saying, just that ending with this sentence is strange. The focus of the article is statistical analysis of exposome studies, not so much on the need for them.

Round 2

Reviewer 1 Report

The authors have modified the manuscript according to the previous comments and all the questions have been well answered.